# Incarcerated Loved Ones: Building a Community to Support and Advocate on Facebook

**DOI:** 10.3390/ijerph20054002

**Published:** 2023-02-23

**Authors:** Eman Tadros, Sarah Presley, Eunice Gomez

**Affiliations:** 1Division of Psychology and Counseling, Governors State University, University Park, IL 60441, USA; 2College of Social Work, The Ohio State University, Columbus, OH 43210, USA; 3College of Education, Northeastern Illinois University, Chicago, IL 60625, USA

**Keywords:** thematic analysis, qualitative, incarceration, incarcerated loved ones, Facebook

## Abstract

Individuals with an incarcerated loved one are often overlooked when discussing the impacts of incarceration. It can be difficult for these individuals to navigate the criminal justice system, as well as form meaningful connections and obtain support from others that are experiencing a similar situation. Social media allows for connections to be made between individuals in similar situations that might not be geographically close to one another. Specifically, for those with an incarcerated loved one, the Facebook group “Incarcerated Loved Ones” allows for meaningful connection to others who are navigating incarceration. Posts were collected from this Facebook group, with the themes emerging including COVID, information seeking, and advocacy. Findings, as well as future directions, will be discussed.

## 1. Impacts of Incarceration on Family

Incarceration is a substantial element of society that continues to lead to negative outcomes for families and extended communities. Families of those incarcerated that are integrated and allowed to connect with their incarcerated loved ones record more positive outlooks [1,2]. However, the literature surrounding the impacts of incarceration often include negative outcomes for children experiencing parental incarceration, such as increased family financial burdens, increased class division, racial disparities, destabilizing family transitions, and pronounced health complications [1,2,3,4]. It has been found that incarcerated individuals suffer from mental health conditions at a much higher rate than the general population [5]. The available literature showcases societal implications that call for a need for criminal justice rehabilitation innovation, particularly for transitioning formerly incarcerated individuals back home and their families. The shared experiences of those with incarcerated loved ones must be listened to in order to understand what these individuals need, in order to enact changes that provide assistance during and after incarceration.

The impact of incarceration is felt long after incarceration periods have ended, as even shorter imprisonment stays have been highly impactful to families [1,6]. Collateral consequences of incarceration are experienced by the incarcerated individual and their connected families [6,7]. Negative outcomes include increased levels of isolation and loneliness [8], as well as familial incarceration leading to the dissolution of marriages and the fracturing of families, as incarceration is seen to increase divorce rates [9]. Barriers due to incarceration affect families and the surrounding community, which entails extended family members and caretakers for children often being left behind [10]. Stigma is a particularly damaging social barrier as it leads to the loss of relationships, an important element due to the vital service a family support network provides to incarcerated impacted families [1,2,11]. For children, the effects of parental incarceration can cause high-risk adolescent behaviors and may eventually induce an onset of social problems, disability, disease, and early mortality when they are adults [12]. Moreover, children with incarcerated parents/guardians are more likely to display behavioral problems at school due to exposure to maltreatment, substance abuse, and violence [13], which may increase the chances for these children to become incarcerated themselves at some point during their lifetime.

The availability of a family support network plays an impactful role in a family’s future post-incarceration; families with communal aid describe positive effects as well as feelings of backing and cooperation; conversely, families without aid from extended family members describe feelings of hopelessness and disparity [2]. These findings suggest that less family restructuring and transitioning help create more stable outcomes for teenage children.

### 1.1. Impacts of COVID on Incarcerated Individuals and Loved Ones

As of March 2022, there have been 464.8 million confirmed cases of COVID-19 and 6 million deaths worldwide reported to the World Health Organization [14]. The pandemic was especially challenging for incarcerated individuals. The COVID-19 pandemic has led to higher reports of isolation, relationship challenges, stress, and hindered communication affecting several demographics, including incarcerated individuals [15,16,17,18]. Families with an incarcerated loved one and involvement in relationships are at a higher risk of emotional instability, as navigating emotions within these relationships, such as highs and lows, can be challenging as compared to navigating emotions on one’s own [19]. Incarcerated populations are vulnerable as there are elevated challenges in the form of severe disorders (bipolar, depression, schizophrenia) and social barriers (financial, employment, housing insecurities) as compared to the general population; these vulnerabilities are compounded by COVID-19 and the impact that the pandemic had on this population [20].

Existing challenges for incarcerated individuals and their families regarding communication have been further strained by the COVID-19 pandemic, as in-person visitation for incarcerated individuals was canceled as the pandemic surged in 2020 [17]. Long distance communication proved to be a barrier with incarceration rules and policies, including fees associated with calling incarcerated individuals, further contributing to loss of communication [17]. It is critical that family members who have incarcerated loved ones have access to support as they cope with the pressures and difficulties brought on by their loved ones’ incarceration. Social support can be divided into four different groups and is characterized as the functional component of relationships. The four support categories are: emotional support, instrumental support, informational support, and appraisal support [21]. To provide emotional support, one must demonstrate empathy, love, trust, and care. Further, giving someone in need tangible assistance and services is known as instrumental support [21]. Informational support consists of giving someone guidance, recommendations, and knowledge they can use to solve problems. A key component of appraisal assistance is the distribution of information that can be used for self-evaluation or constructive criticism and affirmation [21]. Social support groups are an example of community-based involvement since they give people who are going through similar difficulties a place to talk, ask questions, and offer support to one another. The greater cooperation and resource sharing makes the community feel stronger.

### 1.2. Current Study

This study aims to understand the social support that is given, shared, and received within the Facebook group “Incarcerated Loved Ones.” This study seeks to show how each of four support categories are present within this group, but to also understand the implications of the support within this online Facebook group. Qualitative analysis as a whole provides an approach toward uncovering comprehensive socio-contextual summaries of research questions, with qualitative thematic analysis serving as a research tool to establish recurring themes and accurate accounts of data [22,23,24,25,26,27]. Qualitative methods are intricate and diverse, as thematic analysis is a base model for qualitative methods, although it is not always recognized as such [24,28]. Research based on qualitative methods is an essential form of study, however, the intricacy of qualitative research requires in-depth methodological systems to come to concrete conclusions [29]. Utilizing these forms of social support, such as social media, loved ones with an incarcerated individual are able to receive and share this support with others experiencing similar situations.

## 2. Method

### 2.1. Analytic Strategy: Thematic Analysis

The qualitative methodology chosen for this study was thematic analysis. The thematic analysis method highlights the importance and organization of specific themes [27]. Thematic analysis can be used to analyze data and identify representative themes [24]. Themes in data can identify and give meaning to patterns or sequences that correlate to research questions [20]. Thematic analysis excels at evaluating and condensing extensive qualitative data sets to specific points giving rise to communicable themes [28]. Thematic analysis has been used to analyze data from Twitter being distilled down to key themes and insights in the process regarding their research question [30]. It was found that social media communication could provide exceptional insights using thorough qualitative thematic analysis [30].

### 2.2. Participants

The data for this paper was collected from an online Facebook group, “Incarcerated Loved Ones”, which consists of 16.5 k members. These members have all self-identified as a loved one to an incarcerated individual. Anyone can deem themselves a loved one, it is a term that is left open and the group administrator does not discriminate based on relationship to or with the incarcerated individual. The mission of the group, which is directly stated on their page, is to provide encouragement to others in similar situations, advocate for reform within the criminal justice system, and share information among members related to the criminal justice system. One must obtain permission from the group administrator to enter the group by answering the following questions: (1) Do you agree to read & follow the rules? And (2) Do you have an incarcerated loved one and if not why do you wish to join? This is a closed group, indicating that all members must agree to the following five rules: being kind and courteous, no promoting or selling, respecting the privacy of members, no profanity, and no asking for money.

### 2.3. Data Collection Procedures

This study was approved by the Governors State University’s Institutional Review Board as well as given approval by the group administrator. The dataset was derived from individual posts from the Facebook group “Incarcerated Loved Ones” during the time period of 20 July 2021 to 20 September 2021. This time frame was chosen due to the recency of the posts, as data collection began in that period and lasted for two full months. Using this timeframe allows for data to be collected at different periods of time, meaning there would be a variation in the individuals posting as well as the content. Typically, new members post immediately upon being admitted into the group, due to often feeling a sense of panic as they navigate the difficulties surrounding incarceration. The total number of posts over this time period was 862. For this paper specifically, *n =* 441. We excluded posts that included solely images and/or emojis. If a post had an emoji we didn’t eliminate that post, only if the post had simply emojis or simply an image. We did not code comments, only the original post. We also excluded posts that discussed rules and regulations of the group, which were posted by admin such as, “Hello Members, We appreciate that you want to invite your friends to our page, but when you invite your friends to our pages, please be sure to let them know they must answer the membership questions. Over 60 new requests were denied this morning due to not answering the questions. Thank you”.

### 2.4. Data Analysis

The individual posts were copied into an excel spreadsheet, including the initials of the person who posted it and the date of the post. If the post was an image or picture, this was noted in the excel spreadsheet and then described what it consisted of. Once all the data posts were pulled from that time period, the posts were coded in terms of ways group members show support for each other or seek support from one another. The researchers then compiled themes based on the codes that appeared.

To compare and combine notes, modify differences [22,23], and become practical with laws and regulations, authors created a log or codebook. The accuracy of interpretations were checked by a second and third coder to guard against taking any particular narrative out of context. The developed codes/themes are then reviewed by the second coder. To ensure confidentiality, posts containing identifying information with an individual’s name embedded into the post, were removed in addition to the name of the individual posting. 

## 3. Results

In exploring the posts that individuals made within the Facebook group, “Incarcerated Loved Ones”, there were numerous themes that were identified based on the content included. The five overarching themes that emerged from the total *n* = 862, were information seeking, sharing experiences and seeking connection, motivational and inspirational, COVID, and advocacy. However, for the purposes of this study, we will focus on discussing three out of the five themes, with these three themes totaling the final *n* = 441 for this paper. For the purposes of this paper the three themes we will focus on are (1) COVID, (2) information seeking, (3) advocacy (Table 1).

### 3.1. COVID

The COVID-19 pandemic disproportionately impacted the criminal justice system within the United States. For those with a loved one incarcerated, this was a terrifying time, as the virus spread rampantly within facilities. Additionally, court was often delayed during this time, which left individuals waiting in jail for long periods of time. Navigating this pandemic was especially difficult for those who were unable to have updates on their incarcerated loved one, as visitation was suspended in most states and virtual visits were not always available. The codes for this theme include visitation, vaccines, lockdown and outbreak updates, and home confinement.

#### 3.1.1. Visitation

COVID-19 disrupted visitation between incarcerated individuals and their loved ones, as in-person visitation was paused due to the spread of the pandemic. Additionally, virtual visitation was not available in all states or facilities, which made communicating with those incarcerated extremely difficult during this time. Group members shared their experiences regarding visitation during COVID-19, as well as asked questions of the group in order to best navigate this challenge. One group member shared, “First visit in almost 2 years finally on Saturday! We are traveling 5 1/2 h across state lines to see him! Only sad thing is it’s a no contact visit and they are only allowing 2 h due to COVID”!

Another group member shared their experience regarding visitation.


*Happy Saturday! Had my first visit with my fiancé since he’s been locked up 13 days ago. Plexiglass and masked up. SMH Since I was the only one there, we both pulled our masks down. Damn that. Ya know? I did not drive over an hour, go through 2 tolls (one way) to look at my mans face with a mask! Especially when it was just the 2 of us!!!*


Another individual shared,


*He’s allowed 2 visits a week-30 min. If we actually started at my appointment time which was 10:15, then the CO must’ve been feeling generous because when I got to my car, it was 11:10. But it FLEW by. Can’t wait to see him again already…*


#### 3.1.2. Vaccines

Once vaccinations were available for those incarcerated, group members began to discuss vaccines and related questions within the group. These questions often provided various dialogue in the comments, as it was a topic that individuals had strong opinions about. One group member shared,


*Update: I sincerely appreciate all the comments but this thread went in a different direction than Was my intention. I was asking for help with who to contact and it evolved more into Vax vs. Non-Vax. Honestly, that’s a personal choice and I respect both. At the time I was frantic and reaching out for help! I did get some good information & I respectively thank you all!!My son just called and said at Big Sandy they are FORCING the vaccine on inmates!! He does NOT want to take the vax. There will be consequences (like being put in the SHU and more) if they do not take it! Can anyone tell me who to contact about this or what action we can take??? He has until Aug 9!!! TYIA.*


Group members also discussed the requirement of vaccination in order for certain options to be available for their incarcerated loved one. One group member posted, “I just called to set up a VIDEO VISIT and was told he is not eligible UNTIL he takes the vaccine. What the heck? (Texas)”. Good news was also shared regarding vaccination, as one member shared, 


*All federal employees meaning all federal prison cos and staff will have to be vaccinated or show proof of testing negative every 2 weeks. This will help our loved ones since most cases were brought in by staff. At my sons facility they are opening up more and more each week. They are hoping by next week or so to have the yard open all day like it used to be. This is good news none of us want to go backwards, it’s not as much about us, but I’m better knowing those lockdowns are few and far between.*


#### 3.1.3. Lockdown and Outbreak Updates

Navigating the pandemic added another layer of challenges related to communication from correctional facilities, as now those with an incarcerated loved one had to stay updated on the outbreak of COVID-19 at facilities. An outbreak impacted visitation, both in-person and on video, as well as if their incarcerated loved one would be in lockdown with limited access to communication. Group members often shared messages or updates from correctional facilities, as there were individuals in the group who had loved ones in the same facilities. One group member posted, “*picture of an email saying no visitation in Phoenix unless inamte is vaccinated. No cost video visits. Caption: “No contact visits for SCI Phoenix today”. Another group member shared, “Alert: Due to a COVID-19 outbreak, visitation is canceled at Keen Mountain Correctional Center for the weekends of September 4–5 and 11–12. Keen Mountain remains a visitation pilot site and will be evaluated for visitation for the weekend of September 18–19”. These alerts also included updates such as, “picture of COVID updates: visitation and volunteer activities cancelled until further notice”. In sharing these alerts and updates, group members assisted other members with finding out information regarding COVID-19 at the facility their loved one may be incarcerated in.

#### 3.1.4. Home Confinement

During the pandemic, individuals that were deemed low-risk were often sentenced to complete their sentence at home through home confinement rather than remain in prison, in an effort to decrease the prison population to manage the spread of COVID-19. There was concern that once the pandemic ended, or was under control, these individuals would be sent back to prison. Group members’ posts regarding home confinement also focused on advocating to keep their loved one at home. Home confinement was specific to the COVID-19 pandemic, as, without this event occurring, home confinement would have never been an option for many incarcerated individuals.

One individual shared,


*Ok y’all this is the time we ALL need to flood the President with messages asking him to not send the inmates that are out on home confinement under the cares act due to covid from BOP back to prison When the pandemic is over. I’m asking you take 5 min to send a message! Our voices will be heard if we thunder clap (which means all send them within the next few days) I’ve seen miracles happen when we do this! Even if your loved one is not affected by this at all your voice still makes a difference! #VoiceForTheVoiceless #CJAdvocates*


Another group member shared, “link shared: For prisoners released due to COVID, return to prison is devastating (USA TODAY)Caption: We must spread awareness & let people know this can’t happen! We have to be a voice for them”.

### 3.2. Information Seeking

The theme of information seeking encompasses the ways in which individuals with an incarcerated loved one often ask the group for information related to the experience of incarceration. There are numerous topics this encompasses, as for those with an incarcerated loved one there is a lot of information that is not shared but is crucial for navigating the system. This included questions that ranged from asking about plea deal advice to questions regarding visitation, as well as sharing information on various lawyers. Additionally, individuals often asked logistical questions about pre-sentence investigations, parole, contacting the various facilities, and re-entry questions. Group members sought out information in a logistical manner with the questions asked.

#### 3.2.1. Sentencing

Group members would often post regarding sentencing questions on behalf of their loved one in the criminal justice system. Sentencing is often a complicated matter and individuals relied on one another to explain the nuances associated with it. One group member posted,


*Anyone’s sentencing date constantly getting moved? First if was September then November and just spoke to his lawyer today he said now it’s around January he went in when I was 3 months pregnant in January. She’s 3 months now I had her in June He’s only met her 3 times through a glass window.. today’s just one of those days where it feels like my world is crashing Down.*


Another group member posted,


*Hello my boyfriend is getting ready for sentencing in November he mentioned he had his psi I believe is what it was called on Tuesday he said next they will contact me possibly do a walk through of our home does anyone have any tips/suggestions to help me get a idea of what’s going to be the process of it. I have anxiety and like to prepare mentally before hand.*


Members also posted state specific questions related to sentencing, as one member inquired, “So I don’t know if anyone can answer this but in Wisconsin if there was a no contact order there the courts when sentencing is there a way to try to get that overturned?” Another group member posted, 


*I was hoping someone could help me understand this. I am a little overwhelmed with other stuff going on and my brain is not functioning correctly. The sentence reads “ 240 months on Count 1 and 160 months on Count 3, to be served consecutively Sentence to be served consecutively to prior federal conviction in 16-CR-73-LRR” so does that mean he finishes his sentence he is on now then started the 240 then starts the 160?*


#### 3.2.2. Federal Questions

Specifically, individuals with a loved one involved in the federal system often had federal-specific questions that differed from other general questions. The federal system is unique in its own manner, with specific guidelines and conditions that do not extend to the rest of the criminal justice system. Federal-specific questions that were asked related to pretrial release, visitation rules related to the federal system and federal questions related to sentencing. One group member posted,


*Question: When does your federal time start? Date of arrest? Sentencing? Does it matter if you were sentenced by the feds then sent back to state? Should this time still count towards the federal sentence? If anyone can give me the actual case law (I believe that’s what it’s called) that this might fall under that would be great! TIA. Another group member asked, “Does anyone know for federal prison camp visitation, what information the inmate has to provide about the visitors ahead of time? Name, address, phone number? DOB? SS?” One member posted, “Federal question: Does the release date on the BOP website mean the release from prison the the halfway house or the release date from halfway house to home?”*


Another group member posted,


*FEDS ONLY…. Babe is waiting to ride out from the county to the Feds, he’s now starting to get nervous for the 1st time in the 2 years we’ve been waiting! WHAT CAN I TELL HIM TO EXPECT? Any advice?... He’s done 2 yrs in county and has NEVER been to prison let alone a FEDERAL PRISON.*


#### 3.2.3. Attorney Information

Those with an incarcerated loved one often ask group members for recommendations related to attorneys, as a quality lawyer is crucial to have when involved with the criminal justice system. These posts were often location specific, as members would ask for recommendations related to their location. Group members shared posts such as, “ Any referrals for good attorneys for vacating sentences? Out of Massachusetts, thanks” and, “Does anyone know a good criminal attorney in Oklahoma? I need one so bad”. Another member posted,


*HELP!!! The attorney who called ME and offered HIS services is not returning ANY of my calls or texts!!!! I sold my (beeping) car to get him his money to represent my fiancé and now I got CRICKETS!!!!! Court is in 11 days- September 28th @9am! And we don’t want no public PRETENDER!!!! Think I’ll be able to secure a lawyer before then!? My man is in Berks County Jail here in Pennsylvania. Anyone know of any criminal attorneys who know their stuff?!? Please LMK ASAP! T.I.A.*


One member of the group shared,


*I need a criminal lawyer that’s not scared or working for Mississippi to take on Mississippi I’m really trying to get my son home it’s been 28 long years I know they railroad my son. I need someone that’s going to take their time and hear me out I know there are loopholes in his case. He was sixteen at the time, Where is rehabilitation, second chance. They gave him life just didn’t put it on paper a plead bargain not to get life. I need help ready to fight.*


#### 3.2.4. Re-Entry

Individuals within the group shared many comments and questions surrounding re-entry for their incarcerated loved one. This included countdowns until they were released, as well as asking logistical questions related to what was needed for the transition into the community post incarceration. One member shared,


*##Days left!!! I’m so stressed out. Idk if I’m gonna make it. Prayers please! This is so hard! We are both stressed. You would think we’d be excited and we are but just the real world and all of the chaos is causing us not to focus on the positive and it just sucks. I can not wait for him to get home! Also I have a question....Did you and your LO ever argue because it was hard for him to understand how things are outside the fences but things got better when he got home?*


In addition, specific advice wanted included how to purchase a vehicle, which companies would hire individuals with a record, housing information, character letters, as well as how to navigate social services for their incarcerated loved one. Another member posted, “Any advice on Landlines for federal LOs coming home? Tried looking for specific info on BOP site, but it isn’t very specific.” Another group member posted, “my LO got transferred to the everglades re entry center in florida. what exactly does re entry center mean? and can anyone tell me about this facility?” This group member inquired, “Anyone here from vegas or Cali how is the jobs like for people who have felons?”

#### 3.2.5. Facility Specific

Many group members had questions regarding the criminal justice system that were specific to the facilities in which their incarcerated loved ones were located. Individuals were often left to have to ask the group members for the answers to these questions, as facilities were not helpful. Specific questions focused on inquiring about how to send items or mail to their loved ones, asking how to report individuals within the prisons, questions regarding commissary, and technology questions.

Multiple group members posted questions regarding how to report a facility for reasons including mistreatment of incarcerated individuals, the condition of the facility including food, treatment of those who were sick, and reporting abuse by correctional officers. One member posted,


*Idk how many people on here from sd but what do we do if we want to report the facility, my son says conditions in there r bad and lile 30 people in one areas, their not really moving them out or get to go outside or even getting their food packs we order them...i need some help how can we as loved ones make a difference or even report the conditions so they atleast get what we order them?*


Another group member inquired,


*Mail question for Federal-Does everything have to be handwritten or can things be typed? I’m getting sentenced in two week, looking at 3–5 years from the sentencing guidelines and want to write a few books that I’ve been working on for awhile. I want to mail in my research, but I’m already at 70+ pages of handwritten notes and my hand is killing me. Would ideally like to type this. Plus with the five page limit per envelope I would be able to fit more info on a types sheet. Thanks in advance for your help, guidance and advice.*


Group members also often inquired about the marriage process, which is often facility-specific. There are numerous guidelines and regulations that can be confusing to navigate, so this topic allowed for individuals to share their own experiences of getting married while their loved one was incarcerated, as well as tips for how to navigate the process. One group member posted, “Anyone know how I can obtain a marriage license while my mans incarcerated? Since he won’t be able to go to city hall in person to apply with me? Or anyone have any information on how to get married while my mans in the feds… the prison hasn’t been helpful at all with giving me information”. Another member shared, “Someone please give me advice on marriage request my bf sending it in mail he’s new in prison and we going to be married and he got months to be in there what do I do after approved like rings? Dress and more before married”.

### 3.3. Advocacy

Group members often engaged in various forms of advocacy through their posts. Advocacy was broad in the multiple ways that it encompassed advocating for their incarcerated loved one. These individuals would often post on behalf of their loved one, such as seeking out employment opportunities for them or asking for advice on behalf of a situation their loved one was facing. This is a form of advocacy itself, as these members were advocating on behalf of their incarcerated loved one. Additionally, advocacy was present through sharing links or posts as well as resources for others navigating the system and news articles. The codes that will be discussed include providing information and sharing news.

#### 3.3.1. Providing Information and Resources

A form of advocacy is sharing information with others and, specifically within this group, members would share various information with one another. These posts included providing information for various facilities, sharing promotional posts that allowed for JPay at reduced rates, sharing petitions and other resources that could assist incarcerated individuals, such as voting information. One group member posted,


*link shared about petition: Please help with my brothers wrongful conviction. Antonio was wrongfully convicted in 1996, he’s spent 24 years in prison. The victim’s next of kin and surviving victim participated in the parole hearing and spoke on his innocence for the first time but they didn’t care. They deemed him a high risk and denied him parole of 7 years*


Another group member shared a resource for those who will be released from prison soon,


*Hi All, I’m a part of HIRE (hub for integration reentry and employment) a non-profit designed to address the unique needs of system impacted individuals coming home. And, bridging gaps of inequities in the criminal justice system. We are having an on line virtual event with Reuben Miller, author of Halfway Home. We will also have speakers who have returned home speaking on their experiences. Please join! And if you think HIRE can help please reach out!*


Resources were also shared among the group, as one member posted, 


** link shared. Caption: “This is an excellent podcast with guest David O’Neil, TX parole attorney. He gives a lot of pointers on what to do to prepare for parole, finding an attorney, etc. Thought I would share it with ya’ll.” Sharing information on the history of the criminal justice system was also a way to engage in advocacy, through educating group members as one posted, “Just a little FYI, The War on Drugs is a phrase used to refer to a government-led initiative that aims to stop illegal drug use, distribution and trade by dramatically increasing prison sentences for both drug dealers and users. The movement started in the 1970s and is still evolving today”.*


#### 3.3.2. Sharing News

Group members engaged in advocacy through the sharing of news, or through links to articles or videos that focus on the criminal justice system and, oftentimes, the inhumane conditions or other problems that were being shared. In sharing news, these members are focused on bringing awareness to these issues. One group member shared, “Have y’all heard about a case worker getting arrested w 10 lbs of drugs? A captain and 3 CO’s were terminated today at SMCI”. Another group member posted, “*link shared on how prisons are increasingly banning physical mail”. Caption: “It’s truly out of control.” Members often posted regarding conditions as facilities, such as this group member sharing, 


** link attached with secretary info. Caption “State of MD-ECI correctional.. on the East coast where is humid and hot- haven’t had water since Sunday around 6 AM. Still no water today.. they brought in portable potties and have them on “bathroom break schedules.” They can’t wash themselves or their clothes/sheets. They won’t allow them rec time to use the microwave or the phones. They don’t want them calling their families. My LO was able to sneak on the phone and call his mom to tell her. They are giving them 2 bottles of water per person today. How is this acceptable?! They don’t know when it’ll be corrected. We’ll be writing all day tomorrow to everyone in the state. If you’re willing, flood the secretary’s email-.*


Another group member posted, 


*The Associate Warden at the Federal Bureau of Prisons Metropolitan Detention Center in Brooklyn, NY has been charged with killing her husband after she shot him in the face in their New Jersey home. Her arrest is the latest black eye for the Federal Bureau of Prisons, which has been besieged for years by chronic violence, mismanagement and a severe staffing crisis…. [link provided].*


## 4. Discussion

This study was conducted to gain insight into how individuals with an incarcerated loved one utilize online support groups, as this was identified as a gap in the literature. Creating meaningful connections and support with others in the same situation can be difficult when dealing with the criminal justice system, however social media provides a way for these individuals to connect. The findings present that all four categories of social support (emotional support, instrumental support, informational support, and appraisal support) [21] are present within this Facebook group. The themes that will be discussed are (1) COVID, (2) information seeking, and (3) advocacy.

### 4.1. COVID

In this time of COVID-19, video calls provide a way for loved ones to stay connected, and also provide another possible form of communication in the future for those that have difficulties visiting in-person [31]. Forms of communication often include email, face-to-face visits, or phone calls [32]. For the incarcerated, in-person visits can assist with maintaining strong ties with loved ones while incarcerated, as well as increasing rates of support upon release [33]. Increasing social ties is vital, as loneliness has arisen as a key outcome of the COVID-19 pandemic, being experienced by all ages and demographics [34]. Forms of communication between incarcerated individuals and their families or partners, such as telephone calls or letters, have high response rates, however children specifically calling on the telephone or writing back only happened about half of the time [32]. However, due to the COVID-19 pandemic, in-person visits have been suspended at this time. The use of video calls for those incarcerated and their loved ones provide a way to maintain connectedness in a more personable way than a phone call, for children, especially, it allows for eye contact and physical communication through gestures or displaying items [33].

Challenges to the incarcerated community continue outside of incarceration; as the COVID-19 pandemic has accelerated the release of the incarcerated through early release considerations to curtail the spread of the virus, adding to the number of those transitioning out of prison. According to Novisky et al. [35] stated that incarcerated individuals face many risks, including greater risk for self-harm and depression due to lack of social contact and communication with loved ones. This is particularly concerning due to possible social erosion and diminished public support to the incarcerated community during the pandemic due to stigma-influenced beliefs [1]. Additionally, formerly incarcerated individuals re-joining the community are welcomed to a pandemic changed landscape necessitating changes to transition programs [36].

Health care and mental health services changed due to the COVID-19 pandemic, not only for prison populations, but for the general public as well, with the popularization of telehealth. Telemental health (THM) went from a specialized form of treatment method to a standard form treatment, as many in-person visitations went online. Telehealth can benefit from remote services from a wide range of professions, including psychologists, psychiatrists, social workers, and marriage and family therapists (MFTs) [31]. Further changes are needed as gaps in medical data have been identified as a result of the pandemic; a majority of prisons and jails were not reporting medical data, which included whether or not staff and the incarcerated were tested for COVID-19 [35]. Additionally, the COVID-19 pandemic further exposed the vulnerability of prison populations to contagious diseases as infection rates for the incarcerated were up to 5.5 higher than the overall U.S population due to overcrowding, inadequate ventilation, and joint lavatories [35,37].

Due to the disruptions of the COVID-19 pandemic further impacting the at-risk incarceration community, communication has suffered significantly, revealing barriers to connection [38]. A hybrid support model suggests that having both in-person and telehealth would serve to alleviate incarceration risks and restraints [20,24]. This dual support model (in-person and telehealth) could be used by having intake and follow-up remotely, and more thorough examinations and procedures be held in-person. Having further remote support networks following up with those incarcerated would further serve to provide rehabilitation especially in overcrowded facilities [39,40]. Social networks could extend out to the families of the incarcerated to keep them informed as many group members suffered several questions and doubts related to sentencing of their loved ones. Incarceration facilities were found to have little to no help for families of the incarcerated, which becomes more apparent during rapidly changing situations such as the COVID-19 pandemic. In relation to COVID-19, studies have shown that social support among the general population can mitigate the association between COVID-19 anxiety and psychological well-being [41,42]. By connecting people who have a family member who is incarcerated to outside support organizations, as well as other local and national organizations that support children and families of the incarcerated, correctional facilities can help promote more social and emotional support. The Children’s Bureau of the U.S. Department of Health & Human Services, for instance, has a list of these organizations that can be distributed to prisons across the country [43]. In order to promote better interpersonal support, it may also be essential for correctional facilities to reopen visitation options and do so in a way that allows people to interact with a loved one who is incarcerated. According to research, families that visited during a loved one’s incarceration have better mental health measures and are more likely to stay together following release [44].

### 4.2. Information Seeking

Those with an incarcerated loved one are often a population left behind in being provided assistance with navigating the criminal justice system. The nuances and barriers of incarceration can be overwhelming, and these individuals often have to rely on others with similar experiences in order to obtain information regarding the process. Within the Facebook group, individuals sought information regarding a multitude of topics related to incarceration, focused on logical questions related to sentencing, attorneys, parole, bail, as well as the federal system. By providing information to one another and sharing information amongst themselves, a sense of community forms. This sense of community is especially important for this population, as incarceration is often stigmatized and shamed, which makes it difficult to find quality support. This social support from others is crucial in providing assistance to those incarcerated, especially as these loved ones assist with navigating the re-entry process [5,11]. Sharing resources, asking questions and also sharing posts about one’s day were other ways in which support was shown within the group. Group members utilized comments and reactions within Facebook to show support on posts.

To assist the challenges of navigating the criminal justice system, correctional facilities, as well as courts should create fact sheets for families that explain the procedures, as well as answer commonly asked questions. This will assist individuals as they navigate what can often be a confusing time. Additionally, enlisting facilities to provide support to individuals with an incarcerated loved one during this time, through measures such as family programming, discounted phone calls, and other ways to increase engagement, makes sure that the bonds remain strong between incarcerated individuals and their loved ones. This has shown to reduce recidivism rates, as well as increased rates of feeling supported and not isolated during the transition from incarceration to the community [45].

### 4.3. Advocacy

Group members were focused on advocacy related to their incarcerated loved one in a multitude of avenues. They would often post on behalf of their loved one, such as seeking out employment opportunities for them or asking for advice on behalf of a situation their loved one was facing. This is a form of advocacy itself, as these members were advocating on behalf of their incarcerated loved one. Additionally, members were focused on reform related to sentencing as well as prison conditions, and brought awareness to these issues through sharing links to news articles or petitions.

Advocating on behalf of their incarcerated loved one has a positive effect for both the incarcerated individual as well as the individual advocating, as it builds community and a sense of purpose while showing support to those incarcerated. Additionally, those who advocate are supporting one another and creating their own connections among individuals with similar goals. However, group members were limited with how they could express themselves, as posts that included swearing as well as asking for money or sharing GoFundMe’s were removed and those users were blocked. Advocating for places in which individuals can speak freely is important, as self-expression of authentic emotions allows for connection among individuals. Additionally, advocacy includes sharing links such as GoFundMe, and in trying to prevent this, the admin within the group are preventing a form of advocacy from being shared.

Family support offers the incarcerated adult and the non-incarcerated spouse various social and psychological benefits [39] as it is a form of advocacy by inquiring on behalf of another individual. For those who are released, social support or lack of it can tremendously impact the likelihood of recidivism [40]. The types of social support for formerly incarcerated individuals include family, professionals within the field of treatment, romantic partners, or even peers [6,39]. However, due to the strains of incarceration on the family unit, the relationships with family members are often damaged once the individual is released and is not always a source of support.

These barriers have been partially mitigated by advocacy from groups such as Support4Families, which work to better integrate transitioning individuals out of incarceration [46]. Such support provides greater outcomes to at-risk populations that include the attached families and communities. This intervention aims to build manageable and realistic resources within the family unit, as well as long-term resilience for families and their loved ones with incarceration histories. Additionally, by assisting family members in identifying specific resources in their communities and social networks, as well as ways to get immediate help after their loved ones’ release. Support4Families also have ways to identify help in the future after the intervention. The key components identified by Support4Families are reduced stress and strain, better social support, and increased family cohesion [46].

### 4.4. Limitations

One of the limitations in this study is that this form of support is only available to individuals that have Internet access, a Facebook account, and are specific members to this group. Those who are active members of this group are sharing their experiences, asking questions, and overall creating an environment where they can be heard and therefore are more willing to be engaged in support with one another. However, those who are of low socioeconomic status may have little to no Internet access, which can limit their perspectives to the group and make them feel further underrepresented. Additionally, we are uncertain of the demographic information (gender, race, socioeconomic status, and other demographic information) of participants. There is no way to determine the gender, education level, socioeconomic condition, race/ethnicity, and other demographic characteristics of the participants. Knowing the demographic information would have provided a more robust analysis of whose voices are represented in the group.

For this study, codes were developed using thematic analysis. Thematic analysis is multifaceted, but the quality of data can be inconsistent and lack coherence when creating themes from research data [28]. The Facebook posts were deidentified so the participants’ demographic information was unidentifiable in order to protect the person behind the data. Another downside is that Facebook limited the sample size as we only were able to gain the perspective of those that (1) have a Facebook account and (2) are a part of this particular support group. Thus, we could have provided potentially different results if we looked at the overall population.

Excluding posts that included images and solely emojis limited the overall data, in terms of making it consistent. Photographs and emojis can be used to express oneself and make online communication more emotive. By removing this, there is the possibility that the authors may have misinterpreted the significance of what the Facebook group wrote in their posts. Potentially including the images and emojis could have resulted in a different message being shared.

Among the many rules that were set before entering the group, the main one was to not use profanity. One of the admins stated, “Don’t come at me and argue about why you can’t use the F word or other cussing on our pages especially on a Sunday I will block you! This is social media & no one will set the tone for my day! Have some respect & etiquette!!!*”* Swearing is an expression of strong feelings. It is predicted to happen in instances where a strong emotion is expressed or when a person shows a particularly strong attitude toward another individual [47]. However, by banning swearing, it may help members of the group feel comfortable and safe within the group, as this encourages individuals to speak in a respectful manner. This a feature of the data source, not a limitation of the study, but worth stating as the impacts of incarceration can leave loved ones vulnerable and blocking members further isolates and refrains individuals from expressing how they truly feel.

### 4.5. Future Directions

With more data available on the spread of COVID-19 and associated risk factors, mental health and transitioning services must regularly revise the needs of these at-risk populations; with possible avenues including greater telehealth use in combination with in-person visitation [17]. As research progresses and more data becomes readily available the long term effects of the COVID-19 pandemic and its institutional impact will become more apparent, allowing for greater research connectivity. Further research can detail barriers to transition into society for the currently and formerly incarcerated, as collateral consequences have revealed the impact to extend out to connected families and communities [1]. Additional training and internship placement sites for counselors are encouraged to prepare trainees to work effectively with the incarcerated population and their loved ones [48,49].

Furthermore, present data collected from Facebook included what was thought to be mostly female posters, based on profile pictures and usernames; obtaining greater proportions of males may reveal stronger research results. Future research can search for possible stigmas or obstacles for men that may explain lower participation in social support networks; additionally, men may face a greater avoidance in asking for help. Facebook has proved to be a vital source of data in unearthing links and themes when applying research questions and thematic analysis; however, other social networks, such as Twitter, could provide additional data for researchers and should be explored.

## Figures and Tables

**Table 1 ijerph-20-04002-t001:** Codes and Themes.

Themes	
COVID	Visitation, Vaccines
	Lockdown and Outbreak
	Updates, Home Confinement
Information Seeking	Sentencing, Federal
	Questions, Attorney
	Information, Re-entry
	Facility Specific
Advocacy	Providing Information and Resources
	Sharing News

## Data Availability

The data is not available as they are part of a Facebook group where posters have their names and private information that they may not want shared.

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
