# Peer review of "Incarcerated Loved Ones: Building a Community to Support and Advocate on Facebook"

_ijerph, 2023, doi:10.3390/ijerph20054002_

Round 1

Reviewer 1 Report

This is great project! This type of naturalistic inquiry that explicates the real time questions and concerns of families is important and insightful. I have a number of recommendations for strengthening the paper:

The use of language and terminology is an important tool for conveying the values and perspectives of the authors. Person-first language humanizes and preserves the dignity of people who are the subjects of scholarly writing. Some of the terminology you use to refer to people who are incarcerated includes: "the formerly incarcerated" "those incarcerated" "the incarcerated" "incarcerated individuals" "a loved one incarcerated" "incarcerated loved one".  I recommend selecting and using a person-first term and using it consistently throughout the paper.  "incarcerated individuals" or 'an incarcerated individual'  as well as 'families with an incarcerated member' or 'families with an incarcerated loved one' all preserve the humanity and dignity of the people who are the subject of your paper. 

The information in the introduction is all relevant though limited. There is substantial literature out there on the effects of incarceration on loved ones and on the family unit. A more robust background section is needed to highlight the range of existing literature and to position the current project as filling a gap in current knowledge about family concerns.  Section 1.2 provides multiple aims/purposes for the research.  This needs to be organized more coherently with an overall aim and/or series of objectives. 

The methods section will need significant re-organization and revision.

- The content in section 2.4 should appear at the start of the Methods section as the initial statement and description of the methodology for the project. 

- With this type of research, where data is extracted from written text, there are no 'participants'. Instead, you have a data source. Section 2.1 provides a good description of the data source. The authors need to include a description of the steps they took to obtain access to the Facebook group (the data source) including the answers they provided to obtain permission to access the group for the purpose of conducting research. There should also be a statement regarding ethics approval to conduct the research. If ethics approval was not required, this should be noted along with the reason why not. Where it is stated that there are "16.5K members nationwide", it should indicate which nation. Due to the significant systemic, political and cultural differences of criminal legal/correctional systems around the world (not to mention COVID-19 responses), the it is critical to identify which country's correctional system the Facebook group discusses. If there is international membership discussing experiences in several countries, this too should be noted. The section should be re-named 'Data Source'. 

- section 2.2 does describe data collection. Some of the language needs to be re-framed to reflect the extraction of written data from the data source (rather than data collected from participants). Eg: "862 posts by 441 distinct members of the group between July 20, 2021 - September 20, 2021".  The first two sentences of section 2.3 also belong in this section.  I also recommend removing any reference to 'themes' from the data collection section. Unless these themes were established a priori and  provided criteria for the selection of posts to be included in the study, reference to them does not belong in this section.

- section 2.3 (titled Data Collection) should, in fact, be titled Data Analysis, and begin with how the posts were coded, and emerging themes identified, in accordance with the methodology of thematic analysis.   In the final sentence of this section, it is unclear if the whole post was removed or if just the name/identifier was removed.  It seems irregular to remove an entire post, thereby losing important data, if removal of the name can deidentify the group member.

- in the first paragraph of the Results section, the authors refer to 'five overarching themes' then 'four themes were discussed' and earlier in the paper and abstract there is reference to 'three themes' and only three were presented in this paper.  The three themes discussed in the paper are not also part of the five listed. This confusion and incoherence must be cleaned up.

- The results presented were clear and the quotes complemented and added to the reader's understanding. All direct quotes from the data should be within quotation marks. A single quote should not span two paragraphs. 

- in the Discussion, the themes should be discussed in the same order in which they were presented in the Results section.  The Discussion seems to mostly summarize the information provided in the Results section. This section should discuss how the Results inform the overall purpose/aim of the project and/or the specific objectives of the project. Recommendations for change to social services and or prison services should be clearly identified as such in their own sub-section of the Discussion. 

- In the Limitations section, the three sentences are not a limitation of the study... it is a feature that defines the Facebook group membership and, as such, should be in the Methods section that describes the Data Source.   Similarly, the last paragraph about the use of profanity is a feature of the data source, not a limitation of the study. 

The results would not differ if you knew the demographic info of group members - knowing the demographic info would have provided a more robust analysis of whose voices are represented in the group. This IS a limitation of the study. 

In the second paragraph of the Limitation section you state that "Facebook limited the sample size and could have provided potentially different results if we looked at the overall population".  If the Facebook application somehow limited your access to the data source, this should be described in the Data Source section.  What do you mean by 'overall population'?  If you were able to include all posts within a particular time frame, were not all voices who posted within that time frame represented within the data? This needs clarifying. 

Author Response

Dear Reviewer One,

            Thank you for reviewing our manuscript, “Incarcerated loved ones: Building a community to support and advocate on Facebook” for publication into IJERPH. The manuscript has undergone the requested revisions as per both of the reviewers’ feedback.

Reviewer one said, “The use of language and terminology is an important tool for conveying the values and perspectives of the authors. Person-first language humanizes and preserves the dignity of people who are the subjects of scholarly writing. Some of the terminology you use to refer to people who are incarcerated includes: "the formerly incarcerated" "those incarcerated" "the incarcerated" "incarcerated individuals" "a loved one incarcerated" "incarcerated loved one".  I recommend selecting and using a person-first term and using it consistently throughout the paper. "incarcerated individuals" or 'an incarcerated individual'  as well as 'families with an incarcerated member' or 'families with an incarcerated loved one' all preserve the humanity and dignity of the people who are the subject of your paper.” We very much appreciated this comment and worked harder as authors to be one collective voice as well as honor our participants.

Reviewer one said, “The information in the introduction is all relevant though limited. There is substantial literature out there on the effects of incarceration on loved ones and on the family unit. A more robust background section is needed to highlight the range of existing literature and to position the current project as filling a gap in current knowledge about family concerns.  Section 1.2 provides multiple aims/purposes for the research.  This needs to be organized more coherently with an overall aim and/or series of objectives.” This has been reorganized and more information on impacts of incarceration on the family has been added.

         They said, “The methods section will need significant re-organization and revision.

The content in section 2.4 should appear at the start of the Methods section as the initial statement and description of the methodology for the project.” We appreciate this, the sections were reorganized as per reviewer one’s advice. Then they said, “With this type of research, where data is extracted from written text, there are no 'participants'. Instead, you have a data source. Section 2.1 provides a good description of the data source. The authors need to include a description of the steps they took to obtain access to the Facebook group (the data source) including the answers they provided to obtain permission to access the group for the purpose of conducting research. There should also be a statement regarding ethics approval to conduct the research. If ethics approval was not required, this should be noted along with the reason why not. Where it is stated that there are "16.5K members nationwide", it should indicate which nation. Due to the significant systemic, political and cultural differences of criminal legal/correctional systems around the world (not to mention COVID-19 responses), the it is critical to identify which country's correctional system the Facebook group discusses. If there is international membership discussing experiences in several countries, this too should be noted. The section should be re-named 'Data Source'.” Respectfully, we disagree. In multiple other studies we have published we have referred to them as participants. We understand that Facebook and their posts are the data source, but the posters are humans and we do not want to dehumanize them. We thank this reviewer for pointing out that ethics needed to be added and the error about the nation needed to be corrected.

Reviewer one said, “Authors should clarity who loved ones included? (e.g., family, friends, neighbors, romantic partners, kids?)” We appreciate the inquiry. Anyone can deem themselves a loved one, it’s a term that is left open and the group administrator does not discriminate based on relationship to or with the incarcerated individual. Then they said, “ection 2.2 does describe data collection. Some of the language needs to be re-framed to reflect the extraction of written data from the data source (rather than data collected from participants). Eg: "862 posts by 441 distinct members of the group between July 20, 2021 - September 20, 2021".  The first two sentences of section 2.3 also belong in this section.  I also recommend removing any reference to 'themes' from the data collection section. Unless these themes were established a priori and provided criteria for the selection of posts to be included in the study, reference to them does not belong in this section.” We eliminated the mention of the themes as the reviewer is correct, they shouldn’t be mentioned at this point.

Reviewer one stated, “Section 2.3 (titled Data Collection) should, in fact, be titled Data Analysis, and begin with how the posts were coded, and emerging themes identified, in accordance with the methodology of thematic analysis.   In the final sentence of this section, it is unclear if the whole post was removed or if just the name/identifier was removed.  It seems irregular to remove an entire post, thereby losing important data, if removal of the name can deidentify the group member.” The name of the section was an oversight by the authors and we appreciate the reviewer highlighting that for us. We moved the thematic analysis section up so the reader knows the strategy before the analysis. Finally, we clarified the last sentence. Next they stated, “In the first paragraph of the Results section, the authors refer to 'five overarching themes' then 'four themes were discussed' and earlier in the paper and abstract there is reference to 'three themes' and only three were presented in this paper.  The three themes discussed in the paper are not also part of the five listed. This confusion and incoherence must be cleaned up. The results presented were clear and the quotes complemented and added to the reader's understanding. All direct quotes from the data should be within quotation marks. A single quote should not span two paragraphs.” We clarified 5 overarching themes, then mentioned the three that will be discussed here.

We appreciate the compliments as well as insightful feedback and suggestions given to enhance our paper. Taking the reviewer’s feedback strengthened the ideas posited in this paper. We followed the suggestions which was found to greatly improve the paper as a whole. We are so grateful for the chance to revise and improve their work. Thank you for all of your help.

Sincerely,

A Grateful Corresponding Author

Reviewer 2 Report

Summary: 

The authors of this study use qualitative methods to examine themes that emerge from a supportive facebook community for the loved ones of incarcerated people. The study offers insight into the experiences of the loved ones of incarcerated people during the covid-19 pandemic and implications for how to support this population so that the population of incarcerated people can be benefit from the support provided to loved ones.

Introduction

The authors should name the four support categories and include a summary about each in the introduction. The four support categories (emotional support, instrumental support, informational support, and appraisal support) are not mentioned until the discussion. The authors should also focus on making the case as to why focusing on the loved ones of incarcerate people is an important area to explore as well as make connections to previous support efforts that have been studied.

Methods

·      How did the authors elicit information about support given and received? Were comments on post also coded?

·      Was the group and group administrator notified about the study? Did they have an opportunity to consent?

·      Was this study IRB approved?

·      Authors should clarity who loved ones included? (e.g., family, friends, neighbors, romantic partners, kids?)

·      The analytic strategy can benefit from more details about thematic analysis and less about qualitative methods generally. Any support for why qualitative methods should be used might be better suited for the “current study section.”

Results

·      The authors state there are five themes, then four, then three. I would clarify and simplify this.

·      A table was referenced but not included

·      The section on home confinement and advocacy reads as though they are fairly related. I wonder if home confinement can be consolidated with advocacy or support by quotes that make this section unique to this theme alone..

·      A link is shared in line 406-407, but links are not shared when they are referenced elsewhere. I would share all links or no links for consistency.

Discussion

·      Line 425 discusses the formation of sense of community. What about sense of community is important for active members of the facebook group/loved ones of incarcerated people? I would say more about this. 

·      Line 434 – who should provide support? 

·      Line 444 : strong family ties or strong ties with loved ones? Given the focus of this study, I recommend using loved ones consistently throughout (which may include family but doesn’t have to). It appears most of the existing literature is on families. The authors may include that a strength of the study is that this facebook group included many relationships that may not necessarily be family.

Implications:

·      Implications should focus on the loved ones of incarcerated people. The discussion in the covid section of the discussion goes back and forth between implications for incarcerated people (e.g., telehealth) and implications for loved ones of incarcerated people. Focusing on the population studied will provide more clarity. Future studies however may use these findings to understand the experiences of incarcerated people who experience positive support form loved ones.

·      How did facebook limit the sample size? There was no discussion of posts with photos and emojis being excluded in the methods.

·      While swearing can be an expression of feelings, I wonder how much it helped to support feelings safety within the group? A two sided critique of this in the limitation section can be included.

General editorial comments:

Some editing for clarity is needed throughout. Some sentences required this reviewer to read it over more than once to understand what the writer intended to say. Additionally, in the methods section, some past tense words are needed in the methods (e.g., authors created* a log book). Finally, the use of quotations is inconsistent. Some of the direct quotes from posts have quotation marks and others do not. Review of the manuscript with an editor is recommended. 
